# Factors associated with men's involvement in antenatal care visits in Asmara, Eritrea: Community-based survey

**Ghirmay Ghebreigziabher Beraki**[1]*, **Hagos Ahmed**[2], **Aster Michael**[3], **Bereket Ghide**[3], **Bereket Tekie Meles**[3], **Bersabeh Tekle Tesfatsion**[3], **Rida Abdulwahab**[3]

**1** Department of Nursing, Orotta College of Medicine and Health Sciences, Asmara, Eritrea, **2** National Statistics Office, Asmara, Eritrea, **3** Ministry of Health, Asmara, Eritrea

* gberaki83@gmail.com

## Abstract

### Background

Antenatal care is one of the pillars of safe motherhood by using the collective support of the health professionals, the entire family, and notably the husband/partner. Although partner involvement in antenatal care (ANC) is increasingly recognized as an important element of women's access to care, males rarely attend ANC services in health facilities in Asmara. Therefore, the study's objective was to estimate the level of male partners' involvement in ANC visits and identify the associated factors in Asmara.

### Methods

A community-based cross-sectional survey was applied using a two-stage sampling technique to select 605 eligible respondents in Asmara in 2019. Data was collected using a pre-tested structured questionnaire. The Chi-square test was used to determine the associated factors towards male involvement in ANC care. Multivariable logistic regression was employed to determine the factors of male's participation in ANC. A P-value less than 0.05 was considered statistically significant.

### Results

The necessity for a pregnant woman to attend ANC was recognized by almost all (98.7%) of the male partners; however, 26.6% identified a minimum frequency of ANC visits. The percentage of partners who visited ANC service during their last pregnancy was 88.6%. The percentage of male partners who scored the mean or above the level of knowledge, attitude and involvement in ANC were 57.0, 57.5, and 58.7, respectively. Religion (p = 0.006, AOR = 1.91, 95% CI 1.20–3.03), level of education (p = 0.027, AOR = 1.96, 95% CI 1.08–3.57), and level of knowledge (p<0.001, AOR = 3.80, 95% CI 2.46–5.87) were significantly associated factors of male involvement in ANC.

**Data Availability Statement:** All relevant data are within the manuscript and its Supporting Information files.

**Funding:** The author(s) received no specific funding for this work.

**Competing interests:** The authors have declared that no competing interests exist.

**Abbreviations:** ANC, Antenatal Care/Clinic; HIV, Human Immunodeficiency Virus; SPSS, Statistical Package for Social Sciences; WHO, World Health Organization.

## Conclusions

Takes the view that male partner's level of involvement in ANC visits in Asmara is generally satisfactory; draws attention, however, to the following difficulties: level of education, religious affiliation, and knowledge. Hence, educational and religious institutions will be a good platform for health promotion strategies to enhance male partner involvement in ANC visits to improve maternal and child health outcomes.

## Introduction

Recently, there has been a focus on engaging male partners in sexual and reproductive health (SRH) [1]. As early as 1994, at the International Conference on Population Development, it was recognized that men had a crucial role in advancing SRH [2]. The conference marked a paradigm shift in empowering women to control their fertility and have access to safe childbearing by encouraging the engagement of their partners [1]. Since then, the focus on addressing the engagement of men has become more conceptualized in the context of SRH programs [3, 4]

The maternal mortality rate is an indicator of maternal health services to measure the public health status of a country [5]. The World Health Organization (WHO) estimated that globally 295,000 women die annually as a result of complications of pregnancy and childbirth. Sub-Saharan Africa and Southern Asia accounted for approximately 86% of the estimated global maternal deaths, with sub-Saharan Africa alone accounting for roughly 66% [6]. Eritrea's estimated maternal mortality ratio remains unacceptably high (480/100,000 live births) [6]. Therefore, to reduce the high mortality rates in developing countries like Eritrea, men's involvement is crucial to achieving Sustainable Development Goals, with a target of 70 per 100,000 live births in 2030 [7].

In low- and middle-income countries, the participation of men in ANC services has been found to contribute considerable health benefits for women and children [1, 8]. Moreover, researchers have found that male involvement in maternal health is important to increase the uptake of prenatal care, alleviate unnecessary delay in seeking obstetric care, decrease skipping testing for HIV of his partner, and reduce stress and risk-taking behaviours of couples during pregnancy and beyond [8–13]. Therefore, men's involvement in maternal health services is a health promotion strategy to improve maternal and child health [8, 12, 14]. However, especially in sub-Saharan African countries, pregnancy and childbirth have continued to be viewed as solely a woman's issue, despite men having tremendous control over social and economic power over their partners [15]. Additionally, low male involvement in maternal healthcare services remains a problem for healthcare providers in many African countries due to poor access, stigma, and confidentiality of services that were 'unfriendly' to men [16].

Studies have recommended that there is a need for further research which could help to explain the country-specific findings for strengthening measures to capture men's knowledge, attitude, and behaviour related to reproductive, maternal, and child health [17]. In support of this recommendation, systematic review studies also showed that the effectiveness of interventions to encourage men to increase their attitude and knowledge in ANC resulted in more positive maternal and new-born health outcomes [8]. However, regarding the authors' level of knowledge, no published sources regarding household-based surveys around male partners' involvement in ANC are available in Eritrea.

Men's participation in ANC is a form of health behaviour influenced by predisposing, supporting, and driving factors to reinforce behaviour change [18]. Studies indicated that factors reportedly associated with men's involvement in ANC are significantly related to education, employment, income, access to media, number of children alive, ethnicity, religion, waiting time, staff information, and attitude [19–21]. Therefore, this study aims to estimate the prevalence of men's involvement and analyse factors associated with men's involvement in ANC in Asmara, Eritrea.

## Methods

### Study design and period

The study utilized a community-based cross-sectional design to assess the prevalence and associated factors of men's involvement in antenatal care visits in Asmara, Eritrea. The study period extended from April 2019 to June 2019 to get a good representation of the prevalence. A quantitative approach was used.

### Study area and population

The study was conducted in Asmara. It has a population size of 420,408 and 113,413 households, according to the Asmara municipality report of 2018. Asmara is selected by virtue of it being a capital city of Eritrea in which male involvement is currently highly advertised, as a result of which the awareness raisings and their practical applicability need to be gauged numerically.

The city is geographically divided into four administration blocks (clusters): southeast, southwest, northeast, and northwest. Each block has 3–5 sub-administration areas. One sub-administration area was selected by simple random method from each cluster: Akria, Tsetserat, Arbaete-Asmara and Gejeret. The study's target population was Males with pregnant partners or infants residing in Asmara city.

### Sample size

The sample size was determined based on Macfarlane [22], and Daniel [23] theorem using the formula: $n_1 = \frac{Z^2 P(1-P)}{e^2}$ Where n = sample size, z = 95%, C.I = 1.96, P = expected prevalence or proportion and e = precision (in proportion of one; if 5%, d = 0.05).

$$n_1 = \frac{Z^2 P(1-P)}{e^2} \quad n_1 = \frac{(1.96)^2 0.50(1-0.50)}{(0.05)^2} \quad n_1 = 384.16$$

The sample size was adjusted for the size of the population using the following equation:

$$n_2 = \frac{n_1}{N} + \frac{N}{n_1} = \frac{384.16 \times 47798}{47798 + 384.16} = 381.09$$

Since the sampling technique was a multi-stage, a design effect (1.5) was considered then the sample size was adjusted to $n_3 = 571.64$ ($n_3 = deff \ x \ n_2 = 1.5 \ x \ 381.16$). Finally, a 6% non-response rate was added for the final sample size to be n = 605 males. The non-response rate was based on the results obtained by the 2010 Eritrea Population and Health Survey (2010 EPHS) conducted by the National Statistics Office (NSO) of Eritrea, where the rate was found to range from 2.5% to 9.5% for the different regions of the country [24]. The average of this range was taken for this study.

## Sampling procedure and criteria for selection

A two-staged sampling technique was adopted to select the eligible respondents from Asmara. In the first stage, four administration areas, one each from the four clusters of the city, were randomly selected using a simple random sampling method and the overall sample size was proportionally allocated among the four selected administration areas taking into account the number of households in each administration area (Fig 1). List of households with pregnant women was prepared for each administration area and was used as sampling frame for second stage sampling; that is selection of households. The allocated sample of households was selected in each administration area using Systematic Random Sampling (SYRS) method. If more than one eligible partner is found in a selected household, they all qualified for an interview. Men of reproductive age (18–60 years) who had pregnant partners and/or had biological children aged up to 1 year at the time of data collection, had ANC attendant partners, and were willing to participate were included in the study. The 12 months' grace period was employed to prevent recall bias from participants.

## Data collection tool and variable measurement

The questionnaire was developed with reference to a previous similar study [25], as displayed in S1 File. The questionnaire face content validity was assessed using a panel of experts from the Reproductive Health Unit of the Ministry of Health and Midwifery Department in Asmara College of Health Sciences. After evaluating the items, the acceptable items are preserved in the questionnaire, unacceptable items are removed, and modifiable items are revised and corrected by the panel of experts. This was done through cross-checking. This meant that the researcher's content validity measures contained all possible items used to measure the concept. In this study, a valid measure of 0.5 was acceptable, as in the Spearman correlation coefficient. Then, experts in linguistics who are mainly involved in translating translated the questionnaires from English to Tigrigna, the nation's local language. Back translation to English was carried out to check further whether the translations were consistent with originality. Finally, midwives experienced in research evaluated the ANC components in the questionnaire.

The questionnaire was pre-tested on 10% of the sample (61 households) in the Maytemenay administration area of Asmara, which was not selected for the main study before the study period. Estimating the time required to complete a single questionnaire and simplifying the questions that were not easily understood were mainly performed as a result of the pilot study. Finally, the questionnaire was tested for reliability using Cronbach's Alpha statistic, which was estimated to be 0.70, which can be considered acceptable.

Data was collected done through face-to-face interviewers who can speak and understand the language. The modified questionnaire had four main sections. The first section collects socio-demographic characteristics, including age, address, educational level, occupation, marital status, and religion.

The second section was to assess knowledge of ANC with 21 questions consisting of questions related to information about ANC (2 items), ANC visits (4 items), sleep pattern and nutritional advice (5 items), smoking during pregnancy (1 item), danger signs during pregnancy and seeking care (2 items), services provided in ANC (1 item), accompanying the ANC (1 item), birth preparedness (4 items), and place of delivery (1 item). The answer to every item was scored as follows: "correct" (score = 1), and "wrong" (score = 0). This implies that a respondent will have a total of 21 scores if he answered all questions correctly and a total score of 0 if answered none of the questions precisely. Respondents with a total score below the mean were considered to have a poor level of knowledge, and those with a total score greater than or equal to the mean score were considered to have a good level of knowledge of ANC.

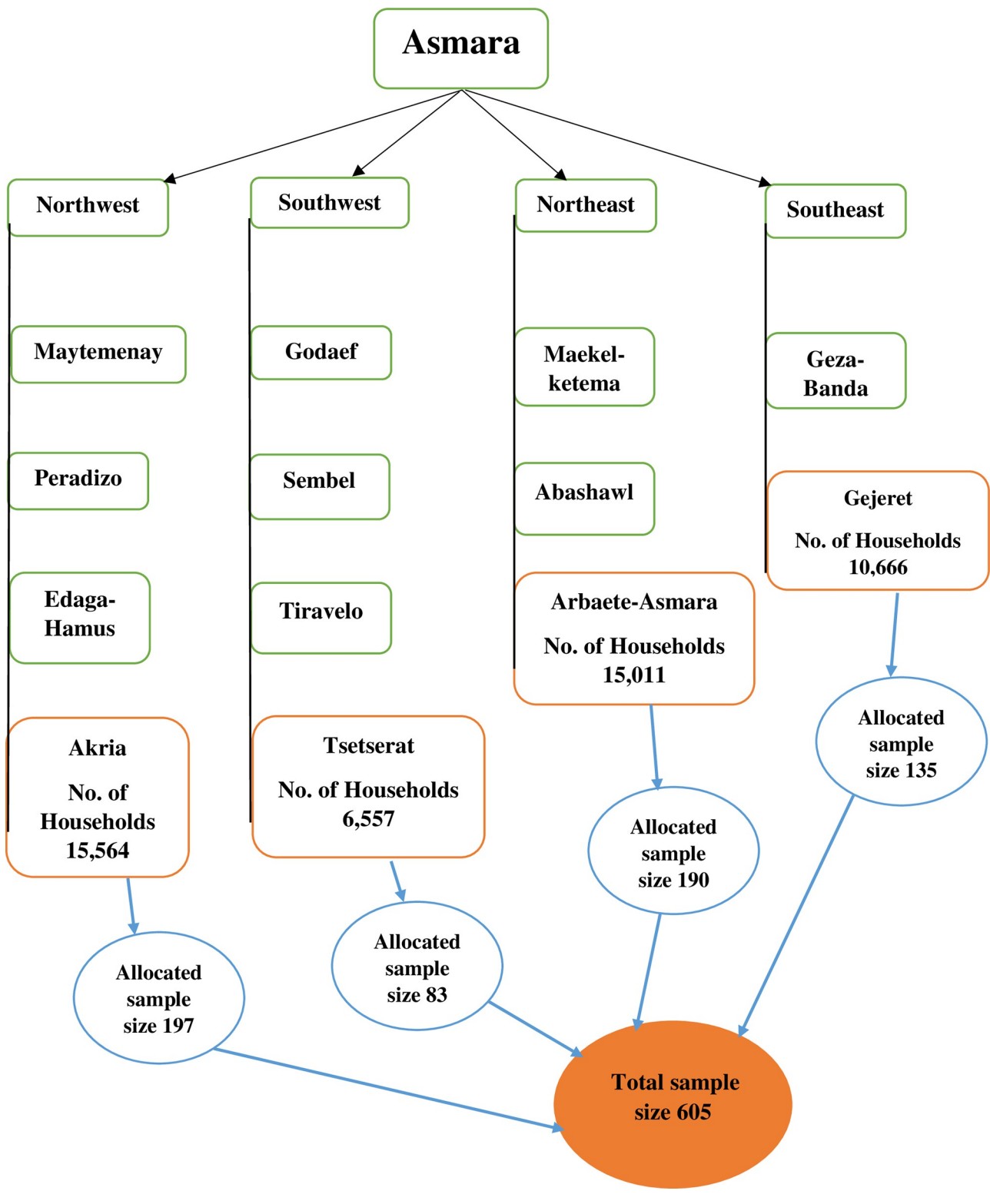

**Fig 1. Flow chart of study site selection and sample allocation.**

The third section assessed attitude towards ANC with 13 questions (each with possible responses of 1 = strongly disagree, 2 = disagree, 3 = medium, 4 = agree, 5 = strongly agree). Questions related to attitude on ANC: attitude on male involvement in accompanying partner to ANC (5 items), time to start ANC (2 items), planning and preparation for ANC (3 items), foodstuff good for the fetus (2 items), and benefit in pregnant woman seeking ANC (1). The total attitude score of respondents was obtained by adding scores to the 13 variables. These imply a respondent will have a total score of 65 if he strongly agrees with all the statements and a total score of 13 if he strongly disagrees with all the statements. Respondents with a total score below the mean were considered to have a negative attitude towards ANC, while the rest with a score greater than or equal to the mean score were considered to have a positive attitude towards ANC.

The fourth section was to assess involvement in ANC with nine questions that included planning of pregnancy (2 items), ANC visits (5 items), and accompanying ANC (2 items), with each question scored as 0 for no involvement and 1 for involvement. The total involvement score was determined for each respondent by adding the responses to the nine questions. This implies a respondent will have a total score of 9 if he had been involved in all the ANC-related statements and a total score of 0 if he does not involve in all the ANC practices. Finally, respondents with total score below the mean score were considered to have a low level of involvement while those more than or equal to the mean score were considered to have a good level of involvement.

## Data entry and analysis

Analysis of the data was performed using SPSS (Version 22.0). The results were summarized using descriptive statistics including absolute numbers and percentages. The significance of the association between the respondents' associated factors and male involvement was carried out using the Chi-square test of independence. Those variables with $p < 0.25$ were further retained for multivariable logistic regression to predict the level of male's involvement in ANC. This helped identify the net impact of each factor on the level of involvement of the respondents in ANC by controlling the effect of other factors in the model.

## Ethics approval and consent to participate

Ethical approval was obtained from the Eritrean Ministry of Health Ethics Committee with a ref. No.: 009/03/18 and research permission was secured from the Orotta College of Medicine and Health Sciences and Asmara Administration Office. The ethical committee approved the procedure for verbal consent for the participants. Then, the study's purpose, nature, potential risk, and benefit were explained to the participants. The participants' rights were protected by ensuring voluntary participation and supported by verbal consent. Anonymity and privacy for the respondents were also maintained throughout, and the respondents were assured that their names would not be divulged at any point of the study. Furthermore, the participants were given a full right to drop from participating or decline to any question in the study, there will be no penalty or loss of benefits to which they are otherwise entitled.

## Results

### Socio-demographic characteristics

This study has a response rate of 94.21% at the initial visit. Participants not present initially were found at the second and third visits by making some convenient appointments, leading to the successful inclusion of all selected partners. The sample characteristics are presented in

**Table 1. Percentage distribution of the respondents by socio-demographic characteristics of (n = 605).**

| Variables | | Number | Percentage |
|---|---|---|---|
| **Age** | | | |
| | 25–34 | 123 | 20.3 |
| | 35–44 | 201 | 33.2 |
| | 45–54 | 182 | 30.1 |
| | 55–59 | 99 | 16.4 |
| **Head of household** | | | |
| | Yes | 558 | 92.2 |
| | No | 47 | 7.8 |
| **Marital status** | | | |
| | Single | 16 | 2.6 |
| | Married | 565 | 93.4 |
| | Cohabiting | 24 | 4.0 |
| **Religion** | | | |
| | Christian | 465 | 76.9 |
| | Muslim | 140 | 23.1 |
| **Level of education** | | | |
| | Illiterate | 3 | 0.5 |
| | Read and Write | 11 | 1.8 |
| | Elementary and Junior (1–8) | 70 | 11.6 |
| | High School (9–12 grade) | 328 | 54.2 |
| | Higher (above 12th grade) | 193 | 31.9 |
| **Respondents current occupation** | | | |
| | Government employee | 443 | 73.3 |
| | Private employee | 54 | 8.9 |
| | Self-employed | 89 | 14.7 |
| | Unemployed | 19 | 3.1 |
| **Partner occupation** | | | |
| | Government employee | 117 | 19.3 |
| | Private employee | 47 | 7.8 |
| | Self-employed | 44 | 7.3 |
| | Unemployed | 397 | 65.6 |

Table 1. The age of the respondents ranged from 25–59 years, with nearly two-thirds aged 35–54 years (63.3%). The majority of the respondents were heads of households (92.2%), married (93.4%), more educated (86.1%), government employees (73.3%), and followed the Christian religion (76.9%).

## Knowledge of the respondents towards ANC

Most (98.7%) of the respondents had heard about ANC, and 61.8% received information from health professionals. Almost all (98.7%) of the respondents understood the necessity for a pregnant woman to attend ANC and were in favour. Still, only 26.6% of them correctly identified the ideal minimum number of ANC visits.

The respondents correctly identified a pregnant woman needs to: take more food (88.8%), sleep for 8–10 hours during nighttime (66.3%) and may require a nap during the daytime (68.6%), and that smoking can harm the unborn child (96.2%). About half (49.1%) of the respondents knew the need for iron and folic acid supplements. However, when they were

asked what time these supplements should be taken, 20.2% answered correctly. Close to three-quarters (73.4%) of the respondents did not understand services rendered during ANC visits except follow-up for pregnant women. Almost all (99.2%) of the respondents identified health facilities as places where one should go in case of emergency; however, only 21.8% of the respondents correctly identified the danger signs of pregnancy. Only 44.8% of the respondents reported at least three of the following importance in accompanying the female partner to ANC: important in obtaining information on both maternal and fetal conditions, improving partners' communication, enhancing male involvement in the pregnancy, moral support, and motivating the male partner to test for HIV.

Respondents identified the need to save money for emergencies during pregnancy (78.3%), arranging transportation for delivery (83.1%), birth preparedness (16.1%), and preparing essential items for clean delivery (18.7%). Most (99%) of the respondents identified the ideal place to have birth as a health facility; however, only 23.8% of the respondents mentioned the need to arrange blood donors in case of delivery complications (S1 Table).

## Attitude of the respondents towards ANC

The statement with the highest percentage (99.2%) of positive attitudes reported by respondents was 'any amount of alcohol drinking during pregnancy will affect the growth of the child'. The second-highest percentage (97.0%) of the positive attitude statement reported by respondents was 'there is a benefit for pregnant women seeking ANC'. Most (95.7%) of the respondents had a positive attitude toward the statement that a male partner should accompany his pregnant partner to ANC for early booking in the first three months; however, 81.7% of the respondents regarded pregnancy as a female domain (S2 Table).

## Male partner involvement in ANC

The majority (78.0%) of the respondents aided in arranging or arranged transportation of their partners to ANC. Almost three of four (76.4%) respondents reported that the last pregnancy was planned. About seven of ten (70.4%) of the respondents participated in the decision made for their partners to seek ANC. Almost three of ten (29.6%) of the respondents did not discuss the topic of pregnancy and childbirth with other people, and 11.4% of respondents reported that their partners had never visited the ANC during their last pregnancy (S3 Table).

## Overall, respondents' knowledge, attitude, and involvement in ANC

As shown in Table 2, 57.0%, 57.5%, and 58.7% of the respondents scored more than or equal to the mean level of knowledge, attitude, and involvement in ANC, respectively.

## Bivariate associates of the level of knowledge and attitude towards ANC

Results in Table 3 show the socio-demographic characteristics considered, level of knowledge about ANC varied significantly by headship status (p = 0.017). Nearly six in ten respondents who were heads of households had a good knowledge of ANC compared to four in ten who were not heads.

Similarly, respondents' socio-demographic characteristics, only marital status was found to have a significant association with the level of attitude of the respondents toward ANC (p = 0.045). Nearly six in ten (58.7%) of the married respondents had a positive attitude towards ANC, compared to only 43.5% of those who were unmarried.

**Table 2. Summary of overall scores on knowledge, attitude, and male partners' involvement in ANC.**

| Variables | | Cut off value | Number | Percentage |
|---|---|---|---|---|
| **Knowledge** | | | | |
| | Below the mean | <13.9 | 260 | 43 |
| | Mean or above | ≥13.9 | 345 | 57 |
| **Attitude** | | | | |
| | Below the mean | <58.5% | 257 | 42.5 |
| | Mean or above | ≥58.5% | 348 | 57.5 |
| **Involvement** | | | | |
| | Below the mean | <6 | 250 | 41.3 |
| | Mean or above | ≥6 | 355 | 58.7 |

Knowledge of ANC significantly contributes to having a positive attitude towards ANC (p = 0.001). Close to two-thirds (63.5%) of the respondents with good knowledge of ANC were found to have a positive attitude towards ANC compared to 49.6% among those with poor knowledge.

**Table 3. Bivariate associates of level of knowledge and attitude towards ANC.**

| Characteristics | | Level of Knowledge | | | Level of Attitude | | |
|---|---|---|---|---|---|---|---|
| | | **Poor** | **Good** | $\chi^2$ **(P-value)** | **Negative** | **Positive** | $\chi^2$ **(P-value)** |
| | | **n(%)** | **n(%)** | | **n (%)** | **n (%)** | |
| **Head of household** | | | | | | | |
| | Yes | 232(41.6) | 326(58.4) | **5.729(0.017)** | 233(41.8) | 325(58.2) | 1.537(0.215) |
| | No | 28(59.6) | 19(40.4) | | 24(51.1) | 23 (48.9) | |
| **Age** | | | | | | | |
| | 25–34 | 60(48.8) | 63(51.2) | 2.823(0.42) | 56 (45.5) | 67 (54.5) | 7.316(0.062) |
| | 35–44 | 79(39.3) | 122(60.7) | | 74 (36.8) | 127(63.2) | |
| | 45–54 | 79(43.4) | 103(56.6) | | 75 (41.2) | 107(58.8) | |
| | 55–59 | 42(42.4) | 57(57.6) | | 52 (52.5) | 47(47.5) | |
| **Marital Status** | | | | | | | |
| | Married | 237(42.4) | 322(57.6) | 1.003(0.317) | 231(41.3) | 328(58.7) | **4.018(0.045)** |
| | *Others | 23(50.0) | 23(50.0) | | 26(56.5) | 20(43.5) | |
| **Religion** | | | | | | | |
| | Christian | 192(41.3) | 273(58.7) | 2.328(0.127) | 196(42.2) | 269(57.8) | 0.089(0.766) |
| | Muslim | 68(48.6) | 72(51.4) | | 61(41.6) | 79(56.4) | |
| **Level of education** | | | | | | | |
| | **Below secondary | 37(44) | 47(56) | 0.238(0.888) | 37 (44) | 47 (56) | 0.779(0.677) |
| | High (Secondary) school | 138(42.1) | 190(57.9) | | 143(43.6) | 185(56.4) | |
| | Higher | 85(44) | 108(56) | | 77(39.9) | 116(60.1) | |
| **Respondents current occupation** | | | | | | | |
| | Government employee | 192(43.3) | 251(56.7) | 0.090(0.764) | 185(41.8) | 258(58.2) | 0.350(0.554) |
| | Non-governmental employees and unemployed | 68 (42.0) | 94(58.0) | | 72(44.4) | 90(55.6) | |
| **Level of Knowledge** | | | | | | | |
| | Poor | | | | 131(50.4) | 129(49.6) | **11.661(0.001)** |
| | Good | | | | 126(36.5) | 219(63.5) | |
| **Total** | | **260** | **345** | | **257** | **348** | |

*Others category under marital status include: single and cohabiting

**Below secondary refers to Illiterate, read and write, elementary and junior level of education

Table 4. Bivariate analysis results of male partners' involvement in ANC.

| Factors | | Poor (n = 250) | Good (n = 355) | χ² (P-value) | Crude Odds Ratio (COR)[95% C.I] |
|---|---|---|---|---|---|
| **Head of Household** | | | | | |
| | Yes | 40.9 | 59.1 | 0.633(0.426) | 1.27[0.70,2.32] |
| | No (Ref.) | 46.8 | 53.2 | | |
| **Age** | | | | | |
| | 25–34 | 35.8 | 64.2 | 7.095(0.069) | 1.50[0.87, 2.57] |
| | 35–44 | 36.8 | 63.2 | | 1.43[0.88,2.33] |
| | 45–54 | 47.8 | 52.2 | | 0.91[0.56,1.49] |
| | 55–59 (Ref.) | 45.5 | 54 | | |
| **Current Marital Status** | | | | | |
| | Married (Ref.) | 41.9 | 58.1 | 0.878(0.349) | 1.35[0.72, 2.53] |
| | Others* | 34.8 | 65.2 | | |
| **Religion** | | | | | |
| | Christian | 38.5 | 61.5 | **6.627(0.010)** | **1.64[1.12,2.41]** |
| | Muslim (Ref.) | 50.7 | 49.3 | | |
| **Level of Education** | | | | | |
| | **Below secondary(Ref.) | 57.1 | 42.9 | **13.580(0.001)** | |
| | High(Secondary) school | 35.7 | 64.3 | | 2.41[1.48,3.92] |
| | Higher | 44 | 56 | | 1.70[1.01,2.84] |
| **Respondents current occupation** | | | | | |
| | Government employee(Ref.) | 40 | 60 | 1.276(0.259) | |
| | Other employees and unemployed | 45.1 | 54.9 | | 1.23[0.86,1.77] |
| **Level of Knowledge** | | | | | |
| | Poor(Ref.) | 58.1 | 41.9 | **52.786(<0.001)** | **3.44[2.45,4.83]** |
| | Good | 28.7 | 71.3 | | |
| **Level of Attitude** | | | | | |
| | Negative (Ref.) | 45.9 | 54.1 | **3.886(0.049)** | **1.39[1.001,1.927]** |
| | Positive | 37.9 | 62.1 | | |
| **Overall Percentage** | | 41.3 | 58.7 | | |

*Others category under marital status include: single and cohabiting

** Below secondary refers to Illiterate, read and write, elementary and junior level of education

### Bivariate and multivariable factors of male partners' involvement in ANC

Tables 4 and 5 signify the association of male involvement with selected socio-demographic characteristics, level of knowledge, and attitude of the respondents. The bivariate analysis results presented in Table 4 revealed that religion (p = 0.010), level of education (p = 0.001), level of knowledge (p<0.001), and level of attitude (p = 0.049) of the respondents were found to be associated significantly with their involvement in ANC.

At the multivariable level, variables that had a p-value less than 0.25 in the bivariate analysis were included due to their potential confounding effect. The results of the multivariate analysis given in Table 5 revealed that religion (p = 0.006), a secondary level of education (p = 0.027), and a good level of knowledge (p<0.000) as significant factors of male involvement in ANC.

Knowledge of ANC was found to have a significant positive contribution toward involvement in ANC. Respondents with good knowledge of ANC were 3.8 times more likely to be involved in ANC compared to those with a poor level of knowledge [AOR: 3.80, 95% CI: 2.46, 5.87]. The odds of being involved in ANC were nearly two times higher among those with a secondary level of education than those who attended below the secondary level of education

Table 5. Multivariate analysis results of male partners' involvement in ANC.

| Variables | | Adjusted Odd Ratio (AOR) | 95% C.I for AOR | P-value |
|---|---|---|---|---|
| **Age in Years** | | | | |
| | 25–34 | 1.56 | [0.78, 3.12] | 0.209 |
| | 35–44 | 1.59 | [0.83, 3.05] | 0.162 |
| | 45–54 | 0.82 | [0.44, 1.51] | 0.522 |
| | 55–59 (Reference) | | | |
| **Religion** | | | | |
| | Christian | 1.91 | [1.20, 3.03] | **0.006** |
| | Muslim (Reference) | | | |
| **Level of Education** | | | | |
| | *Below secondary (Reference) | | | |
| | High (Secondary) school | 1.96 | [1.08, 3.57] | **0.027** |
| | Higher | 1.23 | [0.66, 2.28] | 0.52 |
| **Level of Knowledge** | | | | |
| | Poor (Reference) | | | |
| | Good | 3.80 | [2.46, 5.87] | **<0.001** |
| **Level of Attitude** | | | | |
| | Negative (Reference) | | | |
| | Positive | 1.26 | [0.82, 1.92] | 0.289 |

*Below secondary refers to Illiterate, read and write, elementary and junior level of education

taken as a reference category [OR: 1.96, 95% CI: 1.08 3.57]. The likelihood of participating in ANC was almost two times higher among Christian respondents than among Muslims (OR: 1.91, 95% CI: 1.20, 3.03).

The multivariable analysis result further indicated that the regression model is statistically significant in associating the likelihood of involvement of the respondents based on their level of knowledge, level of education, and religion [$\chi^2$ (5) = 66.99, p<0.001]. The model correctly associated the likelihood of involvement of males in ANC for 79.7% of the respondents.

## Discussion

This study considers the perspectives of male partner involvement in ANC and highlights the constraints and concerns they faced. A remarkable proportion of the respondents with a low level of knowledge (43%) and attitude (42.5%) on ANC are the main concerns and constraints for the level of men involvement (41.3%) based on the results of our study. Similarly, knowledge level of participants was significantly associated, as reported in similar studies in Indonesia (41.2%) [18], Ethiopia (44.4%) [26] and a systemic review study [27]. Hence, to address the constraint, perinatal health education interventions are needed for male partners in all reproductive age categories to improve their knowledge level on ANC.

This insight can help reorient our ANC service to be more need-based and couple oriented than exclusively focused on women in maternal health programs that are inaccessible to men. In the Eritrean health care system, community health agents actively participate in the community to promote a healthy lifestyle in the community [28]. However, about three-fourths of the respondents in this study did not know any ANC services, and this showed that health education programs are not orienting men to share equal responsibilities during pregnancy to improve the maternal health outcome. Besides, most (81.7%) of the study participants also responded that 'pregnancy is a female domain'.

In this study, the percentage of partners with a good level of knowledge was 57.0%, and a similar finding was noted in a study conducted in Iran [29]. However, in our study, only about a quarter (21.8%) of the respondents knew the danger signs of pregnancy in antenatal clinics as compared to that of Nepal (39.3%) [30]. In contrast to our finding, in a study conducted in Ethiopia, slightly more than half of the respondents knew the danger signs of pregnancy [25]. This variation in findings could be due to the questionnaire scoring, the difference in the community's cultural context, level of education, and nature of ANC services rendered in the countries.

Our study demonstrates that the level of knowledge has a significant relationship with male involvement in ANC, similar to the finding of the studies conducted in Myanmar [31] and Tanzania [32]. Moreover, a significant association was also observed between the level of knowledge and attitude of the respondents in ANC. Therefore, it is advisable to invest in knowledge of the husbands/partners on ANC to increase their involvement in ANC.

This cross-sectional study identified that 57.5% of the respondents had a positive attitude towards ANC, which is less than the studies in Ethiopia (73.8%) [25] and Iran (65.3%) [29] but in line with that of Nigeria (56.5%) [33]. The main reason for having about average attitude level of the respondents in this study could be due to the influence of a culture that considers maternal health as a women's domain, lack of services targeting men and lack of privacy at the clinics. Therefore, there is a need to change men's attitudes to have a greater understanding of maternal health care outcomes because men are the main decision-makers of healthcare in families in Eritrea.

The level of involvement in this study (58.7%) is similar to other studies in Ethiopia (59.9%) [25] and Tanzania (56.9%) [32] but higher than the one found in Nepal, which estimated men's involvement at 39.3% [30]. Results in our study also revealed that 53.7% of the respondents accompanied their partners to the antenatal clinic two or more times which is relatively similar to the study conducted in Northern Uganda [32] and higher than in Nepal [30]. Moreover, in our study, 81% of women were accompanied to ANC clinics by their partners at least one time, similar to Ghana (71.9%) [34] but contrary to Bangladesh (47%) [35] and Kenya (68%) [36]. The element that contributed to this high male involvement in ANC in our study could be the introduction of couple counselling and testing for HIV during ANC. Regarding who made the decision to seek health care, in our study, 80.3% of the couples decided on both. Whereas in the study conducted in Ambo Town of Ethiopia, 60.7% of the partners decided alone [25], and in Western Australia, men had no role in decision-making [37].

This study revealed that religion had a strong association with male involvement in ANC, where men of the Christian faith reported relatively higher involvement in ANC than their Muslim counterparts. Studies conducted in Tanzania [12], Nigeria [11], and Cameroon [38] also had similar findings. Therefore, it is advisable to consider religious organizations as a platform to support males to participate more in maternal health. Inconsistent with our findings, many recent reviews and studies stressed how regardless of religious affiliation and /or geographical area, maternal and child healthcare services must also develop new ways of reaching out to men in developing and developed countries [39].

This study has shown that level of involvement was associated with educational level; similarly, in the study conducted in Nepal, academic level and age were associated with male involvement in ANC [30]. Women were more prone to the risk factors associated with unintended pregnancy if the male partner's level of education was low [40]. Conversely, in the study conducted in Eastern Ethiopia, male involvement was proved to have no association with educational level or occupational status but with the wife's occupational status [41]. Almost three fourth of the respondents who were involved in our study reported that the last pregnancy was planned. Several risk factors associated with an intended pregnancy could have

been addressed if the scope of our study had captured enough respondents' data regarding unplanned pregnancy [40]. Therefore, further comprehensive study involving unplanned pregnancies is needed to assess the association of male involvement with unintended pregnancy in Eritrea.

Men's involvement in ANC occurs more when men have good knowledge and positive attitudes toward the health of their partners during ANC [17]. The present study supported these findings, in addition to religion and level of education as significant factors of male involvement in ANC. The current direction of Eritrea's maternal care delivery system is not based on midwifery density versus maternal care services. The understaffing of midwives and inadequacy of infrastructure hinder to provision one-to-one midwifery assistance care protocol in the country. Measuring the adequacy of the midwifery workforce using standards of competency and scope of work is vital to have positive outcome measures in obstetrics [42, 43]. In the absence of such protocol, accommodating male partners in maternity care is challenging [4]. However, Eritrea is striving to alleviate the challenges that have a negative impact on the functionality and resilience of the maternity care practice.

Based on our findings, we recommend that local policymakers and programmer developers use our study's key performance indicators to make male participation in maternal health more effective than ANC's traditional interventions that only targets women in Eritrea [14].

## Strengths and limitations

The survey was capable of estimating the prevalence of men's involvement in ANC visits at community level in Asmara-Eritrea, with an adequate sample size, which can be useful for maternal and child health planners. However, the study cannot determine the causal relationship because of the lack of chronological order of the data collected for the exposure and outcome variables. The participants are also prone to response bias because the replies are recorded as per the self-response of the individuals. It is known that the self-responses are prone to cultural, moral and socially acceptable activities and ideas.

## Conclusions

Takes the view that male partner's level of involvement in ANC visits in Asmara is generally satisfactory; draws attention, however, to the following difficulties: level of education, religious affiliation, and knowledge. Most of the respondents believed that pregnancy is a female domain. This belief can pose a challenge to the involvement of males in ANC. This indicates that there is a need to create an outlook that favours an equal share of responsibilities during pregnancy. Therefore, particular emphasis should be given to male partners who read and write illiterate, elementary and junior level education, higher level of education, and Muslims. Hence, educational and religious institutions can be used as a platform for health promotion strategies to enhance male partner involvement in ANC visits to improve maternal and child health outcomes.

## Supporting information

**S1 Table. Percentage distribution of respondents by knowledge on ANC (n = 605).**
(DOCX)

**S2 Table. Percentage distribution of the respondents by attitude on ANC (n = 605).**
(DOCX)

**S3 Table. Percentage distribution of male involvement in ANC (n = 605).**
(DOCX)

**S1 File. Questionnaire.**
(PDF)

## Acknowledgments

The authors are grateful for the voluntary participation and cooperation of the respondents. We appreciate the support of the administration staff of municipality office of Asmara for allowing us to have access to conduct house to house survey.

## Author Contributions

**Conceptualization:** Ghirmay Ghebreigziabher Beraki, Aster Michael, Bereket Ghide, Bereket Tekie Meles.

**Data curation:** Aster Michael, Bereket Ghide, Bereket Tekie Meles, Bersabeh Tekle Tesfatsion, Rida Abdulwahab.

**Formal analysis:** Ghirmay Ghebreigziabher Beraki, Hagos Ahmed.

**Methodology:** Hagos Ahmed.

**Supervision:** Ghirmay Ghebreigziabher Beraki.

**Validation:** Ghirmay Ghebreigziabher Beraki, Hagos Ahmed, Bersabeh Tekle Tesfatsion, Rida Abdulwahab.

**Writing – original draft:** Ghirmay Ghebreigziabher Beraki.

**Writing – review & editing:** Ghirmay Ghebreigziabher Beraki, Hagos Ahmed, Aster Michael, Bereket Ghide, Bereket Tekie Meles, Bersabeh Tekle Tesfatsion, Rida Abdulwahab.

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
