## [Decision Letter · Decision Letter 0]

10 Sep 2022

PONE-D-22-11568Factors associated with men’s involvement in attending antenatal care visits with their pregnant partners in Asmara, Eritrea: Community based surveyPLOS ONE

Dear Dr. Beraki,

Thank you for submitting your manuscript to PLOS ONE. After careful consideration, we feel that it has merit but does not fully meet PLOS ONE’s publication criteria as it currently stands. Therefore, we invite you to submit a revised version of the manuscript that addresses the points raised during the review process.

Please note that we have only been able to secure a single reviewer to assess your manuscript. We are issuing a decision on your manuscript at this point to prevent further delays in the evaluation of your manuscript. Please be aware that the editor who handles your revised manuscript might find it necessary to invite additional reviewers to assess this work once the revised manuscript is submitted. However, we will aim to proceed on the basis of this single review if possible. Please see the attached file for full reviewer comments. 

We look forward to receiving your revised manuscript.

Kind regards,

Alice Coles-Aldridge

Editorial Office

PLOS ONE

Journal Requirements:

Reviewers' comments:

Reviewer's Responses to Questions

**Comments to the Author**

1. Is the manuscript technically sound, and do the data support the conclusions?

Reviewer #1: Partly

2. Has the statistical analysis been performed appropriately and rigorously? 

Reviewer #1: Yes

3. Have the authors made all data underlying the findings in their manuscript fully available?

Reviewer #1: Yes

4. Is the manuscript presented in an intelligible fashion and written in standard English?

Reviewer #1: Yes

5. Review Comments to the Author

Reviewer #1: It is better to adjust the objectives and title of the study in line with the methods. The article needs editing, which is sent in the attached file. The discussion section should be written more strongly and other results should be compared to yours.

6. PLOS authors have the option to publish the peer review history of their article (what does this mean?). If published, this will include your full peer review and any attached files.

Reviewer #1: No

---

## [Author Response · Author response to Decision Letter 0]

26 Oct 2022

PLOS ONE 

Editorial Office

https://www.editorialmanager.com/pone

Subject: Point by point response to the manuscript PRCH-D-20-02039R3

Dear, Alice Coles-Aldridge 

I, the corresponding author, on behalf of the researchers would like to thank you for your extensive review to our manuscript," Factors associated with men involvement on attending antenatal care visits with their pregnant partners in Asmara, Eritrea: Community based survey "( PONE-D-22-11568). We appreciate your willing to consider our manuscript further once all the concerns have been addressed. Your concerns are addressed as per your recommendations and forwarding to you the responses point by point. 

Yours Sincerely,

Ghirmay Ghebreigziabher Beraki

Corresponding author of the manuscript numbered PONE-D-22-11568

gberaki83@gmail.com

 

Dear Reviewer, 

I, the corresponding author, on behalf of my co-authors would like to forward my appreciation for your constructive suggestions, corrections, comments and questions. Your concerns are addressed point by point. Thank you!

Background

The keywords do not match the mesh term 

Response: Line 43: As per your recommendation, the keywords are modified in the new manuscript as follows. ‘Men involvement, ANC Partner attendant, affecting factors, Asmara’ 

Line 64-69: This paragraph is not consistent with the previous one and they are not related. First, determine the important issue, is the prenatal visits the main issue or the participation of men in the visits? At the beginning of the statement of the problem, these items should be placed 

Response: Line 45-51: Thank you. It is placed as per your recommendation in the first paragraph and the remaining paragraphs are thoroughly revised to address your concern. 

Line 87: If there is no study in your country, but there are studies in other regions about participation barriers and factors related to the lack of participation of men, it is better to use these studies and enrich your problem statement.

Response: Line 81-97: Thank you, your comment well taken and addressed in last paragraph as follows: 

‘Men participation in ANC is a form of health behaviour that is influenced by predisposing, supporting, and driving factors to reinforce behaviour change as suggested by Green et al. (1991) [17]. Studies indicated that factors reportedly associated with men involvement in ANC are significantly related to education, employment, income, access to media, number of children alive, ethnicity, religion, waiting time, staff information, and attitude [18-20]. Therefore, this study aims to assess the level of men involvement and analyse factors associated with men involvement in ANC in Asmara, Eritrea.’

Methods

Line 127: It had no other inclusion and exclusion criteria

Response: Line 122-125: It has inclusion and exclusion and it is restated as follows in the main manuscript. ‘Men of reproductive age (18-60 years) who had pregnant partners and/or had biological children aged up to 1 year at the time of data collection, had ANC attendant partner and willing to participate were included in the study. The 12 months’ grace period was employed to prevent recall bias from participants.’ 

Line 143: Quantitative content validity was not determined by CVI , CVR? 

Response: Line 131-137: The quantitative content validity using content validity index (CVI) and content validity ratio (CVR) were not computed, instead, face content validity was assessed using a panel of experts of reproductive health from Asmara College of Medicine and Health Sciences as well as the Ministry of Health of the country. After evaluating the items, the acceptable items are preserved in the questionnaire, unacceptable items are removed, and modifiable items are revised and corrected by the panel of experts. This was done through cross checking. This meant that the researcher content validity measures contained all possible items that were used in measuring the concept. In this study, a valid measure of 0.5 was acceptable as in Spearman correlation coefficient.

Line 148: Did you have any questions about how to take care during pregnancy, the number and place of care, and other matters related to pregnancy, such as unwanted pregnancy, etc.?

Response: Line 150- 159: Since the care related, pregnancy related and other like questions are included at the items which are supposed to measure the knowledge of the participant, they were not included at the background information. Actually, instead of ‘background information’, we sensibly recognized to be called as ‘demographic background information’, and hence was amended accordingly at the revised manuscript. 

Line 170: The number of questions related to male participation is very low. Did you have any questions about participating in visits, participating in housework, paying for attending the center, etc.?

Response: Questions about participating in visits and other necessary ones were included in the questionnaire (supplementary Table 3). To be specific, the variables that were included are listed below.

The last pregnancy of my partner was planned

My partner has visited ANC

I ever accompanied my partner to ANC

My partner has visited the ANC during her last pregnancy

I have accompanied my partner to ANC at least two times during her last pregnancy

Both me and my partner decide for her to seek ANC

Both me and partner arrange for her transportation

I escort my partner to ANC

I do discuss about pregnancy and child birth with other people

Line 179-180: This is an extra part. It is better to delete it

Response: As per the comment provided, the extra part is deleted in the new manuscript. 

Line 184: The statement of the problem and the title are written in such a way that you are supposed to get the factors affecting the participation of men such as obstacles and improvement strategies, while in the following, knowledge, attitude and performance are given in the methods and the questionnaires are set accordingly. In this case, the title and statement of the problem and goals should be written in this direction.

Response: Line 180-185: As per the comment provided, necessary amendments are made at the new manuscript in such a way as to have consistency in the statement of the problem and goals of the study as follows: 

‘Significance of association between potentially affecting factors of the respondents and male’s involvement was carried out using Chi square test of independence. Those variables with p<0.25 were further retained for multivariable logistic regression to predict the level of male’s involvement in ANC. This helped in identifying the net impact of each factor on the level of involvement of the respondents in ANC by controlling the effect of other factors in the model.’ 

Lin 190-194: Leave the reference of the definitions.

Response: Your comments are well taken and the operational definition section in revised manuscript is deleted.

Results 

Line 188: Briefly explain the method of doing the work. How did men complete 100% of the questionnaires?

Response: Line 188-190: Thank you. ‘This study has a response rate of 94.21% at the initial visit. Participants not present initially were found at the second and third visits, by making convenient appointment, leading to successful inclusion of all selected partners.’ It is corrected accordingly. 

Discussion

Line 297: This whole paragraph is extra. The discussion section should start with the purpose of the study and then your own findings. 

Response: Comment accepted and the extra paragraph is deleted in the revised manuscript.

Line 303: The results of other studies should be mentioned the way of scoring the questionnaire has caused more people to have low knowledge and attitude

Response: Line 283-290: Thank you, this could be a reason to have low level of knowledge. We incorporated your suggestion in the revised manuscript and other studies are also cited to address your concern as follows. 

‘This study considers the perspectives of male partner involvement in ANC and highlights the constraints and concern faced by them. Remarkable proportion of the respondents with low level of knowledge (43%) and attitude (42.5%) on ANC are the main concerns and constraints for level of men involvement (41.3%) based on the results of our study. Similarly knowledge level of participants was significantly associated as reported in similar studies in Indonesia (41.2 %) [17], Ethiopia (44.4%) [25] and a systemic review study [26]. Hence, to address the constraint perinatal health education interventions is needed to male partners in all reproductive age categories to improve their knowledge level on ANC.’

Line 318: One of the factors involved in the difference in the results of the studies is the questionnaires and how they are scored. 

Response: Line 303-306: Thank you, well taken and incorporated in revised manuscript.

Line 329: The difference in attitude is more on which issue? You should mention these things in the comparison of findings.

Response: Line 312-319: You’re right, it is incorrectly stated. The paragraph is corrected in the revised manuscript as follows: 

‘This cross sectional study identified that 57.5% of the respondents had positive attitude towards ANC which is less than the studies in Ethiopia (73.8%) [24] and Iran (65.3%) [28] but in line with that of Nigeria (56.5%) [32]. The main reason for having about average attitude level of the respondents in this study could be due to the influence of culture that considers maternal health as a women’s domain, lack of services targeting men and lack of privacy at the clinics. Therefore, there is a need to bring a change in men’s attitude to have a greater understanding in maternal health care outcome because men are the main decision-makers of health care in families in Eritrea.’ 

Line 363: According to the way of conducting the study, this goal is not correct and the goal and title of the study should be changed.

Response: Line 349-351: The study used both bivariate and multivariate analysis which identified factors associated with the male involvement in attending ANC with their partners. Sorry for not clearly stated the strength of study. It is corrected to avoid the confusion with the goal and title of the study as follows: ‘Having adequate sample size for the estimation of the male’s involvement in ANC is the major strength of the study. However, as the study is a cross-sectional, there is a difficulty to draw causal inferences.’

With regards,

Ghirmay 

Corresponding Author

---

## [Decision Letter · Decision Letter 1]

3 Jan 2023

PONE-D-22-11568R1Factors associated with men’s involvement in attending antenatal care visits with their pregnant partners in Asmara, Eritrea: Community based surveyPLOS ONE

Dear Ghirmay,

Thank you for submitting your manuscript to PLOS ONE. After careful consideration, we feel that it has merit but does not fully meet PLOS ONE’s publication criteria as it currently stands. Therefore, we invite you to submit a revised version of the manuscript that addresses the points raised during the review process.

Editor's commentFor the title of the topic, I suggest you consider reviewer 2 suggestion

We look forward to receiving your revised manuscript.

Kind regards,

Martin Wiredu Agyekum, PhD

Guest Editor

PLOS ONE

Journal Requirements:

Reviewers' comments:

Reviewer's Responses to Questions

**Comments to the Author**

1. If the authors have adequately addressed your comments raised in a previous round of review and you feel that this manuscript is now acceptable for publication, you may indicate that here to bypass the “Comments to the Author” section, enter your conflict of interest statement in the “Confidential to Editor” section, and submit your "Accept" recommendation.

Reviewer #1: All comments have been addressed

Reviewer #2: (No Response)

Reviewer #3: (No Response)

2. Is the manuscript technically sound, and do the data support the conclusions?

Reviewer #1: Yes

Reviewer #2: Yes

Reviewer #3: Partly

3. Has the statistical analysis been performed appropriately and rigorously? 

Reviewer #1: Yes

Reviewer #2: Yes

Reviewer #3: No

4. Have the authors made all data underlying the findings in their manuscript fully available?

Reviewer #1: Yes

Reviewer #2: Yes

Reviewer #3: Yes

5. Is the manuscript presented in an intelligible fashion and written in standard English?

Reviewer #1: Yes

Reviewer #2: Yes

Reviewer #3: Yes

6. Review Comments to the Author

Reviewer #1: (No Response)

Reviewer #2: Authors must addressed the comments attached particularly the measurement of the study variables. The paper contributes a relevant research knowledge by focusing on male partner involvement in ANC services in Eritrea where literature in this area is scanty as expressed by authors. It would be an innovative work if authors rework on the comments on methodology.

Reviewer #3: Dear authors thank you for wrting this manuscript entitled ‘’ Factors associated with men’s involvement in attending antenatal care visits with their pregnant partners in Asmara, Eritrea: Community based survey’’

I found that this manuscript is well written. However, I have a few comments listed below.

1. Title can be modified: into ‘prevalence of men’s involvement and its associated factor……Think over it.

2. This study will be interesting if it was conducted in a hospital, why did you chose community based study, how do you know the true prevalence of male involvement at the community level?

Abstract

1. How many objectives do you have? Line 24 25 showed that only one objective factors associated with male involvement what about the prevalence of male involvement?

2. What is the prevalence of Male involvement?

3. The authors were listed only p value, it will be better if you add the adjusted odds ratio, CI etc.

4. The conclusion section needs revision, what is below and above secondary levels of education?

5. The authors used predictors, factors associated with, factors affecting…be consistent use one words

Background

1. This section needs extensive revision

2. For instance line 53 to 57, start from WHO, then Eritrea!

Results

1. It is better to say socio demographic characteristics rather than Demographic background information.

2. How many questions used to assess knowledge and attitude? And it will be wonderful if you included each variables used to assess knowledge and attitude using table.

3. Use only one table for bivariate and multivariate analysis. And also I did not see the crude odds ratio?

4. In table four the author stated others, what are others you have to write at the end of the table.

Discussion

It is well written

Strength and limitation of the study needs revisions

7. PLOS authors have the option to publish the peer review history of their article (what does this mean?). If published, this will include your full peer review and any attached files.

Reviewer #1: No

Reviewer #2: **Yes: **David Atombire Adumbire

Reviewer #3: No

---

## [Author Response · Author response to Decision Letter 1]

6 Feb 2023

Martin Wiredu Agyekum, PhD 

Guest Editor 

PLOS ONE 

https://www.editorialmanager.com/pone

Subject: Submission point by point response to the manuscript PONE-D-22-11568R1

Dear, Dr. Martin

Thank you for the extensive review and opportunity to submit a revised version of our manuscript. We appreciate the editor and reviewers’ dedication for the extensive and detailed assessment of the manuscript and thoughtful comments. In responding to the comments, we have made a point-by-point revision of the manuscript to reflect most of the suggestions given and rebuttal for the rejected ones, if any. We have highlighted the changes within the manuscript in track changes. We hope this revised version of the manuscript fully meets PLOS ONE’s publication criteria.

We sincerely thank you for considering this manuscript and look forward to your positive responses.

Best regards,

Ghirmay Ghebreigziabher Beraki, MSc

Assistant Professor, Orotta College of Medicine and Health Sciences 

Corresponding author of the manuscript numbered [PONE-D-22-11568R1]

gberaki83@gmail.com

Response to Editor‘s comment 

Response: Line: 1-2: Thank you, comment is well noted. The title is revised accordingly. ‘Factors associated with men involvement in antenatal care visits in Asmara, Eritrea: Community based survey’. 

 

Response to Reviewer # 2

Dear David, 

On behalf of my co-authors, I would like to forward my appreciation for your constructive suggestions, corrections, comments and questions. I am glad to present the point by point responses stated blow to reflect most of the concerns raised to look forward to your positive feedback. Thank you.

Sincerely, 

Ghirmay 

1. Paper Title: Factors associated with men’s involvement in attending antenatal care visits with their pregnant partners in Asmara, Eritrea: Community based survey.

Reviewer suggested title: Factors associated with men’s involvement in antenatal care visits in Asmara, Eritrea: Community-based survey.

• Response: Line: 1-2: Thank you; it is corrected as per your suggestion. 

“Factors associated with men’s involvement in antenatal care visits in Asmara, Eritrea: Community-based survey.” 

No. Reviewer’s Comment

1. THEORETICAL FOUNDATION/GENERAL INFORMATION

i. Clarity of problems being investigated

 The study investigated male partner involvement in antenatal care visits in Asmara, Eritrea. The study provided a justification why partner involvement in maternal health services helps health promotion strategies to improve maternal and child health. However,

i. The study failed to provide statistics on partner involvement in ANC visits in Eritrea. What is the current statistic of male involvement in Eritrea?

• Response: Line: 35 & Line 256-258: Thank you. Your question is addressed. This study provided male partner level of involvement in ANC visits was 58.7% in Asmara-Eritrea. 

2. THE DESIGN AND METHODOLOGY USED

ii. The study used a cross-sectional design. Results are presented in univariate, bivariate and multivariate analysis, which is very appropriate.

However, the study lacks clarity on the unit of analysis. Did the study sample male respondents for interviews or households? If there were more than one household member whose partners were pregnant, were they qualified for the interview? How were the households of the males selected?

• Response: Household with pregnant woman/women with child/children under one year of age was selected for the study. Male partner residing in the selected households were, however, the unit of analysis of the study. If there were more than one household member whose partners were pregnant, they were all qualified for the interview. Unfortunately, there was no selected household with more than one pregnant woman in the study area during the study period. 

iii. There is a methodological discordance as presented by the authors. Which method did the study actually use? Is it purposive or systematic? More clarity needs to be provided on which stage each method was used.

• Response: Line 119-131: Thank you, comment is well noted. You are right; previously, it seemed that the sampling method used to select the cluster from each administration area looks as if purposive. However, now it is written clearly. 

At the first stage, selection of administration area, simple random sampling was used.

At the second stage, systematic random sampling was used to randomly select households from the list of households who had male partner of pregnant women or infants from the list of the selected administration area. 

iv. The study failed to provide justification for the site selection. What are the main parameters that were considered in the selection of the study site?

• Response: Line: 96-98: You raised very interesting point, thank you. The study site is selected by virtue of it being a capital city of the country in which male involvement is currently highly advertised, as a result of which the awareness raisings and their practical applicability need to be gauged numerically. 

v. The study categorization of knowledge of ANC is problematic and must be reviewed. The study measurement of knowledge as poor and good must be reworked. Additionally, the study claimed that “Respondents with a total score below the mean were considered to have a poor level of knowledge, and those with a total score above the mean were considered to have a good level of knowledge on ANC”. What about the respondents who had an average or mean score? I am thinking that all the men or respondents in Asmara, Eritrea could not be lumped as having poor and good knowledge of ANC, some could have moderate knowledge. 

• Response: Line: 163-165: Thank you for your correction. Even though not specifically stated in the method section of the manuscript, it was stated in Table 2 of the analysis results, we have classified those respondents with total knowledge score equal to the mean as having good level of knowledge on ANC.

Actually, it is true that the categorization of the individuals’ knowledge is very controversial all the time. Sometimes, it might not even be advisable to divide into two as ‘good’ or ‘poor’; other times, it might not be preferable to classify it as ‘good’, ‘moderate’, and ‘poor’. Sometimes, it might not be advisable to use the mean or median, and other times it might not be advisable to use some fixed standardized references for the study setting might differ. 

However, the ‘objective’ of the study is usually used as a rule of thumb to make the classification (either to two or three), which in this case the researchers were mainly focusing to determine factors associated with good knowledge among the men as compared to those individuals having poor knowledge. If the categories are made as three, the factors might not clearly be portrayed to the audience for the analysis will be made using multinomial logistic regression. And most of the time, the results of the multinomial logistic regression are not friendly to the readers. 

vi. Lines 157 to 159 must be reworked. 

• Response: Line: 163-165: Thank you. It is corrected accordingly.

vii. Lines 167 to 169 on the measurement of attitudes toward ANC must be reworked. I do not see where to place respondents who had exactly the mean or average score. I am thinking that all the men or respondents in Asmara, Eritrea could not be lumped as having negative and positive attitudes toward ANC, some could have neutral attitudes as well. 

• Response: Line: 173-175: Thank you for your correction. Even though not specifically stated in the method section of the manuscript, it was stated in Table 2 of the analysis results, we have classified those respondents with total attitude score equal to the mean as having positive attitude on ANC. 

The attitude response classification in two (‘positive and negative’ rather than to three (positive, neutral and negative’) is due to similar reason given as in the above ‘Response iv’.

viii. Lines 172 to 175 should be revised. The total marks score for a respondent if he had been involved in all the ANC-related statements should be more than 7 marks. Again, the measurement of involvement or ANC practices should be revised. The two-level categorization is problematic and does not provide clarity. How about men who have a fair level of involvement?

• Response: Line: 181-183: Thank you for your correction. Even though not specifically stated in the method section of the manuscript, it was stated in Table 2 of the analysis results.

The involvement response classification in two (‘good and poor’) rather than to three (‘good, fair and poor’) is due to similar reason given as in the above ‘Response iv’.

ix. The authors might like to read Bloom BS. (1956) Taxonomy education. New York: David McKay. This could help to firm up the assessment of the overall respondents’ KAP (knowledge, attitude and practices) of ANC level based on the Bloom’s cut-off.

• Response: We appreciate your deep search to enhance our manuscript quality. Thank you for the reference material too. However, due to the reasons given above in ‘Response iv’ the survey used the two cut-off option rather than three (Bloom’s cut-off) to best fit for the objectives and analysis of the study. 

3. ANALYSIS AND PRESENTATION

i. The Level of education variable must be revised to meet the five standard levels of education in Eritrea.

• Response: Yes you are right and as it is stated in Table 1, the five levels of education categories were stated separately. However, in the bivariate and multivariate analysis tables, the first three levels were lumped together into one category “below secondary”. This was mainly to have adequate number of cases required by the bivariate (chi-square) and multivariate (logistic regression) analysis and get reliable results.

ii. The total number of respondents is 605. However, in line 203, two variables Respondent’s current occupation and Partner’s occupation have less than the total sum. Where are the rest of the respondents?

• Response: Line: 212-214: You are right. The unemployed respondents and partners were excluded from the Table 1. But now included and corrected accordingly.

iii. There is no clarity in the multivariate analysis. The authors should consider presenting a separate table for the multivariate analysis. It appears the authors have lumped bivariate and multivariate analysis. Table 4 is a bit confusing to the reader. 

• Response: Line: 297-302: Accepted, Table 4 was split into Table 4a and 4b for bivariate and multivariate analysis results, respectively. 

iv. The chi-square (x2) statistic should be added in the bivariate result table in Table 3. This should be revised.

• Response: Line: 272: Well accepted and revised according to your suggestion.

4. MAJOR FINDINGS FROM THE RESEARCH

i. Religion, educational level and level of ANC knowledge significantly influence partner involvement in ANC visits.

• Response: Thanks and your comment is well noted. 

5. Conclusion Based on Data

i. The conclusion is problematic and self-contradictory. In line 244 and 245, the study presented in Table 2, that 57.0%, 57.5%, and 58.7% of the respondents scored more than or equal to the mean level of knowledge, attitude, and involvement in ANC, respectively. However, in line 368 and 369, the authors wrote that; “the study showed that close to sixty percent of the respondents had a good level of knowledge, attitude, and involvement in ANC.” In another breath, the study concluded in line 39 that, “male partner involvement in ANC was considerably low”. This need to be revised.

• Response: Line: 378-387: Thanks and the conclusion is revised thoroughly.

Male partner level of involvement in ANC visits in Asmara-Eritrea was generally good. Nonetheless, most of the respondents believed that pregnancy is a female domain. This belief can pose a challenge to the involvement of males in ANC. Therefore, there is a need to create an outlook that favors an equal share of responsibilities during pregnancy. 

The associated factors in hindering male involvement in ANC visits in this study were level of education, religious affiliation and level of knowledge. Therefore, special emphasis should be given to male partners who read and write illiterates, elementary and junior level of education, higher level of education, and Muslims. Hence, educational and religious institutions can be used as a platform for health promotion strategy to enhance male partner involvement in ANC visits to improve maternal and child health outcomes. 

6. GENERAL COMMENTS

 On the whole, the paper is a good piece. However, the following must be noted and revised:

 (a) The title of the study should be refined as suggested earlier. 

Response: Thank you, it is correct as per your suggestion. 

 (b) The authors need to firm up the justification for selecting the study site.

Response: Well noted, the selection criteria are thoroughly revised.

 (c) More explanation needs to be provided on the multistage sampling to include how the community level was selected from the sub-administration

Response: Accepted, sampling methods are thoroughly revised and corrected.

Response to Reviewer #3:

Dear authors, thank you for writing this manuscript entitled ‘’ Factors associated with men’s involvement in attending antenatal care visits with their pregnant partners in Asmara, Eritrea: Community based survey’’

Dear Reviewer #3, 

We the authors, appreciate your comprehensive assessment of the manuscript, productive suggestions and encouragement. We are gladly to accept and include your suggestions and comments in the revised manuscript.

With regards, 

Ghirmay

Title 

1. I found that this manuscript is well written. However, I have a few comments listed below. Title can be modified: into ‘prevalence of men’s involvement and its associated factor……Think over it. 

• Response: Line: 1-2: Thank you for your suggestion. Suggestion was also given by the other review and editor regarding the title. Based on your suggestions the title is revised as follows: 

Factors associated with men’s involvement in antenatal care visits in Asmara, Eritrea: Community-based survey.

2. This study will be interesting if it was conducted in a hospital, why did you chose community based study, how do you know the true prevalence of male involvement at the community level? 

• Response: Yes, it could be possible to do the study in health facilities. However, most of the time pregnant women come to ANC visits unaccompanied by their partners according to our observation. Hence, there will be limited chance of getting male partners for interview during the ANC visits. 

Regarding the true prevalence of male involvement at the community level in Asmara, because the sample was randomly selected and representative of all male partners in Asmara city, the estimate we get is reliable and represents the true prevalence with reasonable margin of error of 2% and 95% Confidence Interval of (54.7%, 62.5%). 

Abstract 

1. How many objectives do you have? Line 24 25 showed that only one objective factors associated with male involvement what about the prevalence of male involvement? 

• Response: Line: 23-24: Thank you for identifying the error and the manuscript is revised to integrate your correction. The objects are mainly two. The first one is estimating the level of male partners’ involvement in ANC visits and the second objective is to identify the factors associated with the male involvement.

2. What is the prevalence of Male involvement? 

• Response: Thank you for reminder us the point. 

Line 35: The study estimated the prevalence of 58.7% men involvement in ANC visits with margin of error and 95% confidence interval indicated above.

3. The authors were listed only p value, it will be better if you add the adjusted odds ratio, CI etc. 

• Response: Accepted. It is revised accordingly in the abstract section as well in the results section in Table 4b. 

Abstract, Line: 35-37: Religion (p=0.006, AOR=1.91, 95% CI 1.20-3.03), level of education (p=0.027, AOR=1.96, 95% CI 1.08-3.57), and level of knowledge (p<0.001, AOR=3.80, 95% CI 2.46-5.87) were significant associated factors of male involvement in ANC.

4. The conclusion section needs revision, what is below and above secondary levels of education? 

• Response: Thank you, comment is well noted and revised as follows: 

Line: 41-43: Male partner level of involvement in ANC visits in Asmara was generally good. The associated factors in hindering male involvement in ANC visits were level of education, religious affiliation and level of knowledge. Hence, educational and religious institutions will be a good platform for health promotion strategy to enhance male partner involvement in ANC visits to improve maternal and child health outcomes. 

5. The authors used predictors, factors associated with, factors affecting…be consistent use one words 

• Response: Well taken. It is correct as ‘factors associated with’ throughout the text. 

Background 

1. This section needs extensive revision, for instance line 53 to 57, start from WHO, then Eritrea! 

• Response: Comment well taken, the background is revised thoroughly.

Line: 55-62: The World Health Organization (WHO) estimated that globally 295,000 women die annually as a result of complications of pregnancy and childbirth. Sub-Saharan Africa and Southern Asia accounted for approximately 86% of the estimated global maternal deaths, with sub-Saharan Africa alone accounting for roughly 66% [5]. Eritrea's estimated maternal mortality ratio is still unacceptably high (480/100,000 live births) [5]. Therefore, to reduce the high mortality rates in developing countries like Eritrea, involvement of men is crucial to achieving Sustainable Development Goals, with a target of 70 per 100,000 live births in 2030 [6].

Results 

1. It is better to say socio demographic characteristics rather than Demographic background information. 

• Response: Comment well taken, the manuscript was revised accordingly. 

2. How many questions used to assess knowledge and attitude? And it will be wonderful if you included each variable used to assess knowledge and attitude using table. 

• Response: Line: The knowledge and attitude questions are attached as supplementary information to reduce the number of tables in the manuscript. 

3. Use only one table for bivariate and multivariate analysis. And also I did not see the crude odds ratio? 

• Response: Line: 297-302: Your comment is well taken. Separate tables are presented for bivariate and multivariate analysis results. We included the adjusted odds ratio in the multivariate results (Table 4b) and the crude odds ratio and their 95% confidence interval in Table 4a (bivariate analysis results). The crude odds ratio values are obtained by running bivariate logistic regression analysis. In the bivariate analysis we used chi-square procedure which provides chi-square value and p-value to help us decide which factors to include in the multivariate analysis. 

4. In table four the author stated others, what are others you have to write at the end of the table. 

• Response: Line: 299: Thank you for asking the clarification. Others category under marital status include: single and cohabiting and it is stated as per your recommendation. 

Discussion 

It is well written 

• Response: Thank you! 

Strength and limitation of the study needs revisions

• Response: Your suggestion is well taken and the strength and limitation of the study is revised to address your concern as follows:

Line:370-376: The survey was capable to estimate the prevalence of men involvement in ANC visits at community level in Asmara-Eritrea, with adequate sample size, which can be useful for maternal and child health planners. However, the study cannot determine the causal relationship because of lack of chronological order of the data collected for the exposure and outcome variables. The participants are also prone to response bias in the sense that the replies are recorded as per the self-response of the individuals. It is known that the self-responses are prone to cultural, moral and social acceptable activities and ideas.

---

## [Decision Letter · Decision Letter 2]

12 Apr 2023

PONE-D-22-11568R2Factors associated with men’s involvement in antenatal care visits in Asmara, Eritrea: Community-based surveyPLOS ONE

Dear Dr. Beraki,

Thank you for submitting your manuscript to PLOS ONE. After careful consideration, we feel that it has merit but does not fully meet PLOS ONE’s publication criteria as it currently stands. Therefore, we invite you to submit a revised version of the manuscript that addresses the points raised during the review process.

ACADEMIC EDITOR: Authors should explain well the study design.Authors should ensure that the references conform or the same as Plos One referencing style. They should go through the references list one by one as there are mistakes and typos.

We look forward to receiving your revised manuscript.

Kind regards,

Martin Wiredu Agyekum, PhD

Guest Editor

PLOS ONE

Journal Requirements:

Reviewers' comments:

Reviewer's Responses to Questions

**Comments to the Author**

1. If the authors have adequately addressed your comments raised in a previous round of review and you feel that this manuscript is now acceptable for publication, you may indicate that here to bypass the “Comments to the Author” section, enter your conflict of interest statement in the “Confidential to Editor” section, and submit your "Accept" recommendation.

Reviewer #3: All comments have been addressed

Reviewer #4: (No Response)

Reviewer #5: (No Response)

2. Is the manuscript technically sound, and do the data support the conclusions?

Reviewer #3: Yes

Reviewer #4: Partly

Reviewer #5: Yes

3. Has the statistical analysis been performed appropriately and rigorously? 

Reviewer #3: Yes

Reviewer #4: Yes

Reviewer #5: Yes

4. Have the authors made all data underlying the findings in their manuscript fully available?

Reviewer #3: Yes

Reviewer #4: Yes

Reviewer #5: Yes

5. Is the manuscript presented in an intelligible fashion and written in standard English?

Reviewer #3: Yes

Reviewer #4: No

Reviewer #5: No

6. Review Comments to the Author

Reviewer #3: Dear Authors, They have addressed all my concerns. My last comment would be the authors can write ''Prevalence of men's involvement and its associated factors..''

Reviewer #4: I read with great interest the manuscript, which falls within the aim of this Journal. In my honest opinion, the topic is interesting enough to attract the readers’ attention. Nevertheless, the authors should clarify some points and improve the discussion, as suggested below.

Authors should consider the following recommendations:

- Manuscript should be further revised in order to correct some typos and improve style.

- I recommend to highlight how organization of the delivery room and potential understaffing may significantly affect the possibility to adopt one-to-one midwifery assistance and influence the obstetric outcomes during delivery (authors may refer to: PMID: 27924674; PMID: 32307556).

Reviewer #5: Minor revision

The manuscript deals with a key field in the health education of pregnant women.

The manuscript has received revisions from other colleagues, and the paper is much improved.

• Considering the review comment of other reviewers and the lack of information regarding the number of male household members where the woman was pregnant and the question regarding how the households of the males were selected, the sentence in the conclusion of both the abstract section and the manuscript should be changed as follows:

Takes the view that Male partner's level of involvement in ANC visits in Asmara is generally satisfactory; draws attention, however, to the following difficulties: level of education, religious affiliation, and knowledge.

• Line 366: Please change as follows:

Based on our findings, we recommend that local policymakers and programmer developers use our study's key performance indicators to make male participation in maternal health more effective than ANC's traditional interventions that only target women in Eritrea

• The authors often cited and commented the religious attitudes. Many recent reviews and studies have stressed how regardless of religious affiliation and /or geographical area, maternal and child healthcare services must also develop new ways of reaching out to men in developing and developed countries. Please add a comment in the discussion and the following reference.

Palioura Z, Sarantaki A, Antoniou E, Iliadou M, Dagla M. Fathers' Educational Needs Assessment in Relation to Their Participation in Perinatal Care: A Systematic Review. Healthcare (Basel). 2023 Jan 9;11(2):200. doi: 10.3390/healthcare11020200. PMID: 36673568; PMCID: PMC9859150.

• Almost subjects involved reported that the last pregnancy was planned. If we want to expand educational programs, authors should promote future studies involving unplanned pregnancies. Please add a comment in the discussion or the limitation section and add the following reference.

Seifu CN, Fahey PP, Hailemariam TG, Atlantis E. Association of husbands' education status with unintended pregnancy in their wives in southern Ethiopia: A cross-sectional study. PLoS One. 2020 Jul 9;15(7):e0235675. doi: 10.1371/journal.pone.0235675. PMID: 32645075; PMCID: PMC7347164.

7. PLOS authors have the option to publish the peer review history of their article (what does this mean?). If published, this will include your full peer review and any attached files.

Reviewer #3: No

Reviewer #4: No

Reviewer #5: **Yes: **PAOLA DI CARLO

---

## [Author Response · Author response to Decision Letter 2]

25 May 2023

Martin Wiredu Agyekum, PhD 

Guest Editor 

PLOS ONE 

https://www.editorialmanager.com/pone

Subject: Submission of point-by-point response to the manuscript PONE-D-22-11568R2

Dear, Dr. Martin

Thank you for your further consideration of our manuscript. In response to the comments, we have accomplished a point-by-point revision of the manuscript that reflects the suggestions given. We have highlighted the changes within the manuscript in track changes. We hope this revised version of the manuscript fully meets PLOS ONE’s publication criteria.

We sincerely thank you for considering this manuscript and look forward for your positive responses.

Best regards,

Ghirmay Ghebreigziabher Beraki, MSc

Assistant Professor, Orotta College of Medicine and Health Sciences 

Corresponding author of the manuscript numbered [PONE-D-22-11568R2]

gberaki83@gmail.com

Response to Academic Editor

Authors should explain well the study design.

• Response: Line: 92-95: Thank you, your comments are well noted. The design of the study is explained well to address the gap. ‘The study utilized a community-based cross-sectional design to assess the prevalence and associated factors of men’s involvement in antenatal care visits in Asmara, Eritrea. The study period has extended from April 2019 to June 2019 to get a good representation of the prevalence. A quantitative approach was used.’ 

Authors should ensure that the references have to behave in the same manner as Plos One referencing style. They should go through the references list one by one as there are mistakes and typos

• Response: Your comments are well taken. We adopted the Plos One referencing style in the revised manuscript. The entire manuscript was also checked using grammar checker software for any typos in the entire text of the manuscript. 

 

Dear Reviewer # 3,

Thank you very much for your encouragement and support which was very helpful to improve our manuscript. Our responses are stated below. 

With regards, 

Ghirmay 

Response to Reviewer # 3

They have properly addressed all my concerns. My last comment would be if the authors can write on the ''Prevalence of men's involvement and its associated factors.''

• Response: Your last comment regarding the title of the manuscript is to be reviewed to include ''Prevalence” in the title. As it was mentioned in the previous revision, the title was corrected to ‘Factors associated with men’s involvement in antenatal care visits’ after the comments of the academic editor and two reviewers’. We discussed with my co-authors concerning the retitling again and we agreed to leave the title as it is, trusting that, your concern on the ‘prevalence’ will be addressed without being in the title. Sorry for not reconsidering your comment in the revised manuscript. 

Dear Reviewer #4, 

We the authors, appreciate your great interest in our manuscript. Your suggestions are very helpful and we are so glad to include your points in the revised manuscript.

Best regards,

Ghirmay 

Response to Reviewer #4

Manuscript should be further revised in order to correct some typos and improve style.

• Response: Your comments are well noted. The entire manuscript was checked using grammar checker software to correct the typos in the text.

I recommend to highlight how the organization of the delivery room and potential understaffing may significantly affect the possibility to adopt one-to-one midwifery assistance and influence the obstetric outcomes during delivery (authors may refer to: PMID: 27924674; PMID: 32307556)

• Response: Line 380-386: We read the suggested articles (PMID: 27924674; PMID: 32307556) and found them interesting. Your suggestions are included in the revised manuscript as follows: 

‘The current direction of Eritrea's maternal care delivery system is not based on midwifery density versus maternal care services. The understaffing of midwives and inadequacy of infrastructure hinder to provide one-to-one midwifery assistance care protocol in the country. Measuring the adequacy of the midwifery workforce using standards of competency and scope of work is vital to have positive outcome measures in obstetrics [42, 43]. In the absence of such protocol, accommodating male partners in maternity care is challenging [4].’ 

Dear Reviewer #5, 

We the authors, appreciate your valuable comments and suggestion which are vital to our manuscript quality. Thank you so much! We are so happy to cooperate with your suggestions and recommendations in the revised manuscript point by point as follows:

Response to Reviewer #5 

Considering the review comment of other reviewers and the lack of information regarding the number of male household members where the woman was pregnant and the question regarding how the households of the males were selected, the sentence in the conclusion of both the abstract section and the manuscript should be changed as follows: 

Taking the view that Male partner's level of involvement in ANC visiting in Asmara is generally satisfactory; draws attention, however with all these following difficulties in: the level of education, religious affiliation, and knowledge.

• Response: Line 40-42 & 401-403: Thank you for your concise suggestion to our conclusion section. Your comments are well accepted. The conclusion section is revised in the abstract and main body of the manuscript as per your recommendation as follows: 

‘Takes the view that male partner's level of involvement in ANC visits in Asmara is generally satisfactory; draws attention, however, to the following difficulties: level of education, religious affiliation, and knowledge.’ 

The authors often cited and commented the religious attitudes. Many recent reviews and studies have stressed how regardless of religious affiliation and /or geographical area, maternal and child healthcare services must also develop new ways of reaching out to men in developing and developed countries. Please add a comment in the discussion and the following reference.

Palioura Z, Sarantaki A, Antoniou E, Iliadou M, Dagla M. Fathers' Educational Needs Assessment in Relation to Their Participation in Perinatal Care: A Systematic Review. Healthcare (Basel). 2023 Jan 9;11(2):200. doi:10.3390/healthcare11020200. PMID: 36673568; PMCID: PMC9859150.

• Response: Line 362-365: Your recommendation is well taken and corrected accordingly.

‘Inconsistent with our findings, many recent reviews and studies stressed how regardless of religious affiliation and /or geographical area, maternal and child healthcare services must also develop new ways of reaching out to men in developing and developed countries [39].’

Almost all the subjects involved, reported that the last pregnancy was planned. If we want to expand educational programs, authors should promote future studies involving unplanned pregnancies. Please add a comment in the discussion or the limitation section and add the following reference.

Seifu CN, Fahey PP, Hailemariam TG, Atlantis E. Association of husbands' education status with unintended pregnancy in their wives in southern Ethiopia: A cross-sectional study. PLoS One. 2020 Jul 9;15(7): e0235675.doi: 10.1371/journal.pone.0235675. PMID: 32645075; PMCID: PMC7347164

• Response: Line 368-376: Your comments are well noted and corrected as follows:

‘Women were more prone to the risk factors associated with unintended pregnancy if the male partner’s level of education was low [40]. Conversely, in the study conducted in Eastern Ethiopia, male involvement was proved to have no association with educational level or occupational status but with the wife's occupational status [41]. Almost three fourth of the respondents who were involved in our study, reported that the last pregnancy was planned. Several risk factors associated with an intended pregnancy could have been addressed if the scope of our study had captured enough respondents' data regarding unplanned pregnancy [40]. Therefore, further comprehensive study involving unplanned pregnancies is needed to assess the association of male involvement with unintended pregnancy in Eritrea.’ 

Line 366: Please change as follows:

Based on our findings, we recommend that local policymakers and programmer developers use our study's key performance indicators to make male participation in maternal health more effective than ANC's traditional interventions that only target women in Eritrea

• Response: Line 388-390: Your comments are well noted and corrected accordingly. 

‘Based on our findings, we recommend that local policymakers and programmer developers use our study's key performance indicators to make male participation in maternal health more effective than ANC's traditional interventions that only target women in Eritrea.’

---

## [Decision Letter · Decision Letter 3]

12 Jun 2023

Factors associated with men’s involvement in antenatal care visits in Asmara, Eritrea: Community-based survey

PONE-D-22-11568R3

Dear Dr. Beraki,

We’re pleased to inform you that your manuscript has been judged scientifically suitable for publication and will be formally accepted for publication once it meets all outstanding technical requirements.

Kind regards,

Martin Wiredu Agyekum, PhD

Guest Editor

PLOS ONE

Additional Editor Comments (optional):

Reviewers' comments:

Reviewer's Responses to Questions

**Comments to the Author**

1. If the authors have adequately addressed your comments raised in a previous round of review and you feel that this manuscript is now acceptable for publication, you may indicate that here to bypass the “Comments to the Author” section, enter your conflict of interest statement in the “Confidential to Editor” section, and submit your "Accept" recommendation.

Reviewer #4: All comments have been addressed

Reviewer #5: All comments have been addressed

2. Is the manuscript technically sound, and do the data support the conclusions?

Reviewer #4: Yes

Reviewer #5: Yes

3. Has the statistical analysis been performed appropriately and rigorously? 

Reviewer #4: Yes

Reviewer #5: Yes

4. Have the authors made all data underlying the findings in their manuscript fully available?

Reviewer #4: Yes

Reviewer #5: Yes

5. Is the manuscript presented in an intelligible fashion and written in standard English?

Reviewer #4: Yes

Reviewer #5: Yes

6. Review Comments to the Author

Reviewer #4: I carefully evaluated the revised version of this manuscript.

The authors have performed the required changes, improving significantly the quality of the paper.

Reviewer #5: The manuscript is highly valuable to the scientific community and should be accepted for publication.

7. PLOS authors have the option to publish the peer review history of their article (what does this mean?). If published, this will include your full peer review and any attached files.

Reviewer #4: No

Reviewer #5: **Yes: **Paola Di Carlo

---

## [Editor Report · Acceptance letter]

19 Jun 2023

PONE-D-22-11568R3 

Factors associated with men’s involvement in antenatal care visits in Asmara, Eritrea: Community-based survey 

Dear Dr. Beraki:

I'm pleased to inform you that your manuscript has been deemed suitable for publication in PLOS ONE. Congratulations! Your manuscript is now with our production department. 

Kind regards, 

on behalf of

Dr. Martin Wiredu Agyekum 

Guest Editor

PLOS ONE